# Habitat selection in natural and human-modified landscapes by capybaras (*Hydrochoerus hydrochaeris*), an important host for *Amblyomma sculptum* ticks

**Thiago C. Dias**[1,2,3]*, **Jared A. Stabach**[3], **Qiongyu Huang**[3], **Marcelo B. Labruna**[4], **Peter Leimgruber**[3], **Katia M. P. M. B. Ferraz**[5], **Beatriz Lopes**[5], **Hermes R. Luz**[4,6], **Francisco B. Costa**[6,7], **Hector R. Benatti**[4], **Lucas R. Correa**[2], **Ana M. Nievas**[8], **Patrícia F. Monticelli**[8], **Ubiratan Piovezan**[9,10], **Matias P. J. Szabó**[11], **Daniel M. Aguiar**[12], **José Brites-Neto**[13], **Marcio Port-Carvalho**[14], **Vlamir J. Rocha**[2]*

1 Centro de Ciências Biológicas e da Saúde, Programa de Pós-graduação em Ecologia e Recursos Naturais, Universidade Federal de São Carlos, São Carlos, São Paulo, Brasil, 2 Departamento de Ciências da Natureza, Matemática e Educação, Laboratório de Fauna, Universidade Federal de São Carlos, Araras, São Paulo, Brasil, 3 Conservation Ecology Center, Smithsonian National Zoo & Conservation Biology Institute, Front Royal, Virginia, United States of America, 4 Departamento de Medicina Veterinária Preventiva e Saúde Animal, Faculdade de Medicina Veterinária e Zootecnia, Universidade de São Paulo, São Paulo, São Paulo, Brasil, 5 Departamento de Ciências Florestais, Escola Superior de Agricultura "Luiz de Queiroz", Universidade de São Paulo, Piracicaba, São Paulo, Brasil, 6 Departamento de Patologia, Programa de Pós-graduação em Biotecnologia do Renorbio, Universidade Federal do Maranhão, São Luís, Maranhão, Brasil, 7 Departamento de Patologia, Faculdade de Medicina Veterinária, Universidade Estadual do Maranhão, São Luís, Maranhão, Brasil, 8 Faculdade de Filosofia, Ciências e Letras, Laboratório de Etologia e Bioacústica, Universidade de São Paulo, Ribeirão Preto, São Paulo, Brasil, 9 Embrapa Pantanal, Corumbá, Mato Grosso do Sul, Brasil, 10 Embrapa Tabuleiros Costeiros, Aracaju, Sergipe, Brasil, 11 Faculdade de Medicina Veterinária, Laboratório de Ixodologia, Universidade Federal de Uberlândia, Uberlândia, Minas Gerais, Brasil, 12 Faculdade de Medicina Veterinária, Laboratório de Virologia e Rickettsioses, Universidade Federal do Mato Grosso, Cuiabá, Mato Grosso, Brasil, 13 Programa de Vigilância e Controle de Carrapatos e Escorpiões, Secretaria Municipal de Saúde, Americana, SP, Brasil, 14 Divisão de Florestas e Parques Estaduais, Instituto Florestal, São Paulo, São Paulo, Brasil

* diasthiago93@outlook.com (TCD); vlamirrocha@hotmail.com (VJR)

**Data Availability Statement:** All relevant data are within the manuscript and its Supporting Information files.

## Abstract

Human activities are changing landscape structure and function globally, affecting wildlife space use, and ultimately increasing human-wildlife conflicts and zoonotic disease spread. Capybaras (*Hydrochoerus hydrochaeris*) are linked to conflicts in human-modified land-scapes (e.g. crop damage, vehicle collision), as well as the spread and amplification of Brazilian spotted fever (BSF), the most human-lethal tick-borne disease in the world. Even though it is essential to understand the link between capybaras, ticks and BSF, many knowledge gaps still exist regarding the effects of human disturbance in capybara space use. Here, we analyzed diurnal and nocturnal habitat selection strategies of capybaras across natural and human-modified landscapes using resource selection functions (RSF). Selection for forested habitats was higher across human-modified landscapes, mainly during day-periods, when compared to natural landscapes. Across natural landscapes, capybaras avoided forests during both day- and night periods. Water was consistently selected across both landscapes, during day- and nighttime. Distance to water was also the most important

**Funding:** This study received financial support from the Fundação de Amparo a Pesquisa do Estado de São Paulo (www.fapesp.br; FAPESP grants 2015/04795-3 and 2013/18046-7) and the Fundação Parque Zoológico de São Paulo (www.zoologico.com.br). KMPMBF is funded by Conselho Nacional de Pesquisa e Desenvolvimento Científico e Tecnológico (CNPq) research grant (www.cnpq.br; processes 308503/2014-7 and 308632/2018-4). Funders had no role in the study design, data collection and analysis, decision to publish, or preparation of the manuscript.

**Competing interests:** The authors have declared that no competing interests exist.

variable in predicting capybara habitat selection across natural landscapes. Capybaras showed slightly higher preferences for areas near grasses/shrubs across natural landscapes, and distance to grasses/shrubs was the most important variable in predicting capybara habitat selection across human-modified landscapes. Our results demonstrate human-driven variation in habitat selection strategies by capybaras. This behavioral adjustment across human-modified landscapes may be related to increases in *A. sculptum* density, ultimately affecting BSF.

## Introduction

An increasing number of wild species are being forced to adapt to human-modified landscapes and to live within close proximity to humans [1–3]. Across these landscapes, human disturbance has been altering wildlife distribution [4], behavior [5], activity [3], movement [6], and habitat selection [7]. Mammals, for example, tend to move less and to be more nocturnal in human-modified landscapes [3, 6]. Human influence is also linked to the emergence of almost all zoonosis [8, 9], including tick-borne diseases such as Lyme in the United States [9], Encephalitis in Europe [9], and Brazilian spotted fever (BSF) in Brazil [10]. In that context, obtaining accurate data of wild species in human-modified landscapes, mainly those related at some level to human-wildlife conflict and zoonosis epidemiology, is a challenging and crucial goal to wildlife managers and public health institutions.

Capybaras (*Hydrochoerus hydrochaeris*), the largest living rodents on the planet [11], are distributed across all South American countries, except for Chile [12]. These semi-aquatic grazing mammals are usually found in habitats with arrangements of water sources, forest patches and open areas dominated by grasses [12, 13]. Water is a key resource to capybaras, used for thermoregulation, mate and predator avoidance [12, 14]. Forests provide shelter from the day heat, and a resting place at night [15]. Low herbaceous plants are the main components of capybaras diet [16], and the species has been recorded grazing in open areas [17], where these plants are abundant. Capybaras also show daily variation in habitat use [16], feeding mainly during the day in the Brazilian Pantanal [15] and during the night across human-modified landscapes [16].

Benefited by the great abundance of high-quality food resources from agricultural crops and reduced presence of large predators, capybara populations have recently experienced rapid growth in human-modified landscapes over the last few decades [12, 18, 19]. Over some regions, large populations of capybaras are linked to increased crop damage [20], increased vehicle collisions [21], and the spread of Brazilian spotted fever (BSF)—the most human-lethal spotted fever rickettsiosis in the world [10]. Capybaras are responsible for maintaining and carrying large numbers of *Amblyomma sculptum* ticks, the natural reservoir and main vector of the bacterium *Rickettsia rickettsii*, the etiological agent of BSF [10]. Capybaras can also act as amplifying hosts of *R. rickettsii* among *A. sculptum* populations [10, 22].

The role of vertebrate-amplifying hosts in sustaining *R. rickettsii* populations has been well-discussed, with results showing that *A. sculptum* is unable to sustain the bacterium by itself over consecutive generations [10, 23]. In the Brazilian Cerrado, previous research showed that this tick species is more abundant in forested habitats (cerradão and gallery forests) than in open fields or seasonally flooded habitats [24, 25]. In this context, understanding how capybaras select their habitats across landscapes with different levels of anthropogenic disturbance and vegetation cover (open field versus forests) may have important implications for the ecological relationships between capybaras, ticks, and consequently, BSF.

In this study, we investigated and quantified the variation in diurnal and nocturnal habitat selection strategies by GPS-tracked capybaras across natural and human-modified landscapes. We tested the prediction that capybaras show daily variation in habitat selection preferences across landscapes with different levels of human disturbance, increasing their selection for forests and water sources during daytime periods in human-modified landscapes, to avoid humans.

## Methods

### Ethical statements

Capybara field capture was authorized by the Brazilian Ministry of the Environment (permit SISBIO No. 43259–6), by the São Paulo Forestry Institute (Cotec permit 260108–000.409/2015), and approved by the Institutional Animal Care and Use Committee of the Faculty of Veterinary Medicine of the University of São Paulo (protocol 5948070314).

### Study area

Capybaras were tracked in natural landscapes of Mato Grosso and Mato Grosso do Sul states and across human-modified landscapes of São Paulo state (Fig 1). To assess the level of human disturbance at our study sites, we incorporated the Human Footprint Index (HFI) developed by Venter et al. [26]. This index provides a global map of human pressure in the environment, being useful to assess locations under high levels of human disturbance or areas more likely to be in a natural state [26]. HFI ranges from 0 (natural landscapes) to 50 (high-density built landscapes) and the spatial resolution of the global dataset is 1-km.

Study areas in natural landscapes (São José, Ingá, Ipanema and Poconé) were located in the Pantanal biome. The Pantanal is the largest wetland in the world, characterized by a mosaic of upland vegetation and seasonally flooded areas [14, 30]. This biome consists of large areas of natural vegetation and well-structured/stable ecological communities. The Pantanal supports an extraordinary concentration and abundance of wildlife [31], including a large assemblage of medium and large carnivores [32, 33]. Within the sampled areas of Pantanal, capybaras had no access to crops or exotic grasses.

Unlike natural landscapes, human-modified landscapes in São Paulo state underwent significant land use and cover changes during the second half of the 19th and early 20th century, transforming natural vegetation (Atlantic rainforest and Cerrado biomes) into a mosaic comprised of small forest fragments surrounded by an agro-pastoral matrix [34]. These forest fragments likely experience large edge effects and reduced biodiversity [35], which affects the abundance of medium and large carnivores across the region. Jaguar (*Panthera onca*), puma (*Puma concolor*), anacondas (*Eunectes* spp.), and caimans (*Caiman* spp.) face threats in the state according to the "São Paulo State Redbook of Fauna Threatened by Extinction" [36].

Across human-modified landscapes, we tracked capybaras in six municipalities: Americana, Araras, Piracicaba, Pirassununga, Ribeirão Preto and São Paulo (Fig 1). With the exception of the municipality of São Paulo, all areas were located in agricultural landscapes. Sugar cane, corn, cultivated pasturelands, and small forest fragments were the dominant landscape components in the study sites. In Ribeirão Preto, the area used by capybaras was surrounded by a fence that prevented animals from accessing agricultural crops, but they did have access to exotic grasses, as it was also the case in the other human-modified landscapes. In São Paulo municipality, capybaras were monitored in Alberto Löfgren State Park, a protected area within a forest/urban matrix with plenty of cultivated grasses.

It is important to emphasize that no case of BSF has been reported in Mato Grosso and Mato Grosso do Sul states, and serological analyses of capybaras from these natural landscapes

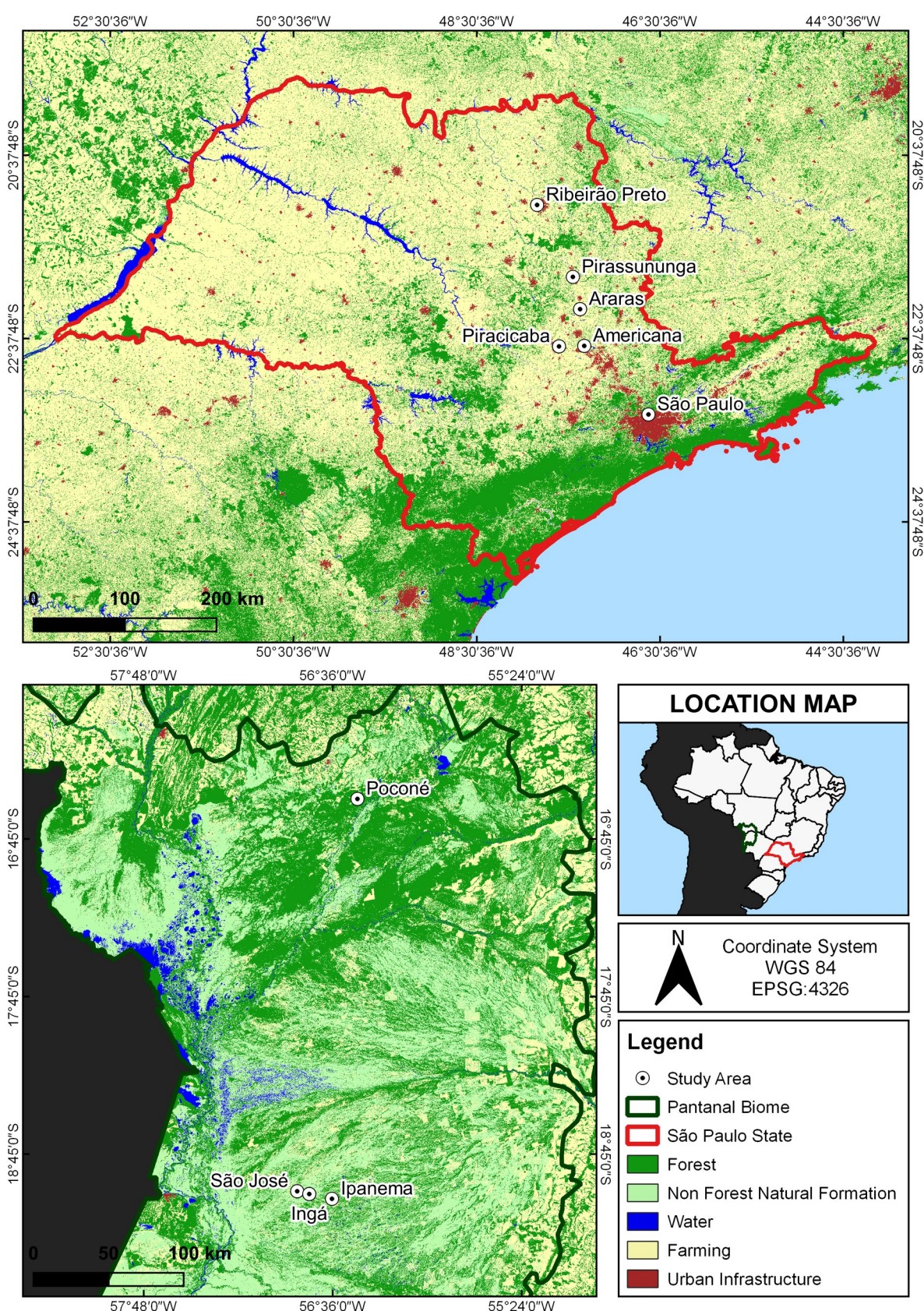

**Fig 1. Study areas across natural and human-modified landscapes in Brazil.** We tracked capybaras from four groups in the Brazilian Pantanal (natural landscapes; green color), located in the states of Mato Grosso and Mato Grosso do Sul, and from seven groups in human-modified landscapes of São Paulo state (red color), in the municipalities of Americana, Araras, Piracicaba, Pirassununga, Ribeirão Preto and São Paulo. Land cover layer was downloaded from Project MapBiomas [27]. Brazilian states shapefile was downloaded from IBGE [28]. South America shapefile was downloaded from Orogénesis Soluciones Geográficas [29]. Geographic Coordinate System: WGS 84 / EPSG 4326.

have shown no evidence of *R. rickettsii* exposure [37]. In contrast, at least three study areas of human-modified landscapes in São Paulo state were classified as BSF-endemic (municipalities of Americana, Araras and Piracicaba), with recent occurrence of human cases and serological evidence of *R. rickettsii* infection in capybaras [37].

## Capybara capture and collaring

From 2015 to 2018, we tracked 20 capybaras from 11 groups in Brazil (S1 Table) with Lotek Iridium Track M 2D GPS collars (Lotek Wireless, Haymarket, Ontario, CN). Among these, four capybaras were tracked from four groups in natural landscapes, and 16 capybaras from seven groups in human-modified landscapes (for more details on tracked individuals see S1 Table). In São José, Ingá and Ipanema ranches (natural landscapes of Brazilian Pantanal in municipality of Corumbá, state of Mato Grosso do Sul), individuals were tranquilized and captured with the aid of a pneumatic rifle (Dan-Inject model JM Standard, Denmark). We used a mixture of ketamine (10 mg/kg) and xylazine (0.2 mg/kg) to anesthetize captured animals [38]. As capybaras use water [11], we targeted animals at a large distance (>20m) from this resource to reduce risk of drowning during tranquilization and capture. Across all other study areas, we captured capybaras through corral-type traps, following the methodology in Pereira & Eston [39].

To better understand movement of capybara populations and minimize the mortality risk of tracked animals, we focused GPS collaring entirely on females. Females show lower agonistic interaction rates when compared to males [40] and therefore, have a decreased chance of mortality. Most female capybara are found in social groups [17, 41] and are thought to be philopatric [42]. We targeted the largest females within each group for GPS collaring because there is a significant correlation between weight and hierarchical position [40]. Hence, we assumed that dominant female movement provided the best representation of group movement.

To avoid incorporating geolocations with large spatial errors [43], we removed GPS positions with a Dilution of Precision (DOP) > 9, following recommendations in Lotek's GPS collaring manual (Lotek Wireless, Haymarket, Ontario, CN.). The day of capture was removed from analyses to reduce bias in space use related to capture-induced stress [44]. Individuals with < 100 data points were also removed. Original GPS-data were collected every 1 or 2-hours during the first 30–40 days, and collars were reprogrammed to collect data every 4-hours and 17 minutes thereafter. GPS-data were rarified until they reached minimum time intervals of 4-hours. Data were categorized into diurnal and nocturnal according to sunrise and sunset time using the '*maptools*' package [45] in the R statistical environment [46]. If a given GPS location was collected between sunrise and sunset, it was classified as diurnal. If collected between sunset and sunrise, the GPS location was classified as nocturnal.

## Habitat data

To assess the level of human disturbance at each study site, and consequently justify the partition of areas into natural and human-modified landscapes, we calculated the mean GPS-data

coordinate of each tracked individual and created a buffer around it with radius equal to the mean dispersal distance of capybaras from their groups (3.4-km) [47]. We then merged buffers of individuals tracked from the same group and calculated the mean HFI within them. These operations were conducted using QGIS 2.18.9 [48]. Across natural landscapes, mean HFI ranged from 2.4 to 6.8 ($\bar{x} = 4.5$; n = 4), and in human-modified landscapes mean HFI ranged from 17.4 to 37.7 ($\bar{x} = 29.2$; n = 7).

To generate covariate data for our habitat selection analysis, we performed a supervised land cover classification using Random Forests, an ensemble learning method common for classifying satellite imagery [49]. We used multispectral high-resolution imagery (2-m resolution) acquired by the WorldView-2 satellite (DigitalGlobe, Inc.) and ancillary data derived from each satellite scene for classification (Table A in S1 Appendix). We established four habitat classes across natural landscapes (forest, water, grasses/shrubs, bare soil) and five in human-modified landscapes (we added a settlements/roads class). The land cover classification was performed using the '*RStoolbox*' package [50] in the R statistical environment [46].

We digitized 1531 training polygons in QGIS 2.18.9 [48] based on visual interpretation of Worldview-2 satellite scenes. Polygons were divided into calibration (70%; used as input for the land cover classification) and validation (30%; used to evaluate the classification). Overall accuracy ranged from 0.95 to 1 in natural landscapes ($\bar{x} = 0.97$; n = 3) and from 0.84 to 0.99 in human-modified landscapes ($\bar{x} = 0.94$; n = 6). We also applied a post-classification filter to reduce 'salt-and-pepper' noise generated by per-pixel classifiers [51]. More details on the land cover classification can be found in S1 Appendix.

For each study area, we calculated the Normalized Difference Vegetation Index (NDVI) [52], and created a binary classification of three habitat layers with ecological relevance to capybaras: forest, water and grasses/shrubs. Forest layers included all the types of forested vegetation, primary or secondary, native or not. Water layers included lakes, ponds, and rivers. Grasses/shrubs layers included native and exotic underbrush and shrubby vegetation, including pasturelands, and agricultural crops.

Using binary habitat classifications, we generated distance layers and calculated the shortest distance between each capybara tracking location and habitat classes. For forest distance calculations, we excluded 50-m from the forest edge to assess selection for areas into the forest interior and edges as well. Large double-lane highways found at some of our study sites (varying from 32 to 44 m width: Rodovia Ernesto Paterniani, Rodovia Luis de Queiroz and Rodovia Anhanguera) likely present barriers to capybara's movement. Because tracked animals did not cross highways during our study, habitats located beyond these highways were not included in our models. Distance to forest interior, distance to water, distance to grasses/shrubs, and NDVI were used as input parameters for resource selection models.

## Resource selection functions

We evaluated habitat selection by comparing the use and availability of habitats through a fine-scale third/fourth-order [53] resource selection function (RSF) analysis [54]. Day and nighttime periods were analyzed separately, due to recognition that capybara habitat use varies throughout the circadian cycle [15]. Habitat availability was determined using a set of random points generated within a buffer around each "use" point (GPS-data) [7, 55]. Buffers were generated with radius sizes equal to the maximum step length displaced by each animal over a time interval equal to our GPS-data resolution (approximately 4-hours). Therefore, each capybara had a unique set of buffers created using its maximum step length in which random points were generated to calculate habitat availability.

To determine the appropriate number of random points per 'use' point (GPS-data), we performed a sensitivity analysis following details described by previous works [7, 55]. We randomly selected one individual from each study area and fit multiple logistic regression models across several possibilities (1, 2, 3, 5, 10, 20, 30 and 50) of random points. We repeated the process 100 times and calculated the expectation of the coefficient estimates and the 95% simulation envelopes. We determined that a sample of 30 availability points per 'use' point provided stable coefficient estimates (Fig A in S2 Appendix). The analysis was performed in R [46].

We included habitat variables in our RSF after determining that they were not highly correlated (Pearson's r > 0.65). To facilitate comparisons across landscapes and across time periods, we scaled and centered all data layers ($[x - \bar{x}]/\sigma_x$). We included quadratic terms for all habitat variables to test for non-linear relationships. Habitat selection was modeled applying a generalized linear mixed-effects logistic regression, following the equation:

$$\omega(x_i) = \exp(\beta + \beta_1 x_{1i} + \ldots + \beta + \beta_n x_{ni} + \gamma_i) \qquad (1)$$

Where $\omega(x_i)$ is the RSF, $\beta_n$ is the coefficient for the $n$th predictor habitat variable $x_n$, and $\gamma$ is the random intercept for the animal $i$. We incorporated random effects into the model structure to better account for differences between individuals, while also accounting for unbalanced sampling designs [56]. We used nested random effects ("individual" inside "study area" inside "landscape") to evaluate landscape-level coefficients. A hierarchical approach was used to account for non-independence between individual movements [7]. Habitat selection was modelled using the 'lme4' package [57] in R [46].

## Models

We created four candidate models (forest, water, open areas and full) for each landscape and time-period (Table 1) and used Akaike's Information Criterion (AIC) to rank them [58]. Models were created to evaluate the importance of different resources on capybara habitat selection: (1) forest—providing shelter from daytime heat and a resting place during the night [15]; (2) water—used by capybaras for thermoregulation, mating and as a refuge from predator attacks [12]; and (3) open areas—used for grazing to meet energy demands [16]. A fourth model, inclusive of all variables, was tested to evaluate if a combination of factors most influenced capybara habitat selection.

We compared all models to a null model using chi-squared tests in R [46]. Coefficients of top-ranked models with confidence intervals that overlap zero were considered statistically insignificant. Top-ranking models were evaluated following the technique in [59], applying Spearman rank correlations between area adjusted frequencies, using presence-only validation predictions and RSF bins (S3 Appendix).

**Table 1. Model structure and number of input variables (K).**

| Model | Structure | K |
|---|---|---|
| Null | - | 3 |
| Forest | Distance to forest interior + (Distance to forest interior)$^2$ | 5 |
| Water | Distance to water + (Distance to water)$^2$ | 5 |
| Open Areas | Distance to grasses/shrubs + (distance to grasses/shrubs)$^2$ | 5 |
| Full | NDVI + (NDVI)$^2$ + Distance to forest interior + (Distance to forest interior)$^2$ + distance to grasses/shrubs + (Distance to grasses/shrubs)$^2$ + Distance to water + (Distance to water)$^2$ | 11 |

## Results

### Capybara capture and collaring

A total of 20 capybaras were captured and fitted with GPS collars. Capybaras were monitored for 33 to 918 days ($\bar{x} = 273$ days), with a similar number of positions collected across study areas (S1 Table). Average fix success was high for both landscapes, ranging from 87% to 99% in natural landscapes ($\bar{x} = 94\%$; $n = 4$) and from 94% to 99% in human-modified landscapes ($\bar{x} = 98\%$; $n = 16$). Maximum distance displaced by individuals in 4-hour time interval ranged from 442-m to 1437-m across natural landscapes ($\bar{x} = 958.2$; $n = 4$) and 268-m to 2703-m in human-modified landscapes ($\bar{x} = 867.6$; $n = 16$).

### Natural landscapes' models

The full model was top-ranked across day- and nighttime periods in natural landscapes, indicating that all habitat variables were important in predicting capybara habitat selection (Table 2). Cross-validation highlighted a strong fit to our data (Table A in S3 Appendix), with stronger results for daytime periods (*day average $r_s$ = 0.83; night average $r_s$ = 0.69*). In natural landscapes, distance to water was the most important variable predicting capybara habitat selection (Table 3), with higher coefficient during nighttime periods (day: β = −1.52±0.03; night: β = −1.91±0.03; Table 3). NDVI was a weak variable in predicting capybara habitat selection during day periods and was not significant during nighttime periods (day: β = 0.21±0.02; night: β = 0±0.02; Table 3).

Capybaras selected areas further from forest interiors in natural landscapes (Fig 2), with highest probabilities of selection found in areas >250-m from the forest centroid (day- and nighttime periods). Capybaras displayed strong preferences for areas near water. This trend was consistent across day- and nighttime periods (Fig 2), with the probability of selection declining with increasing distance. Preferences for areas near open areas, dominated by grasses/shrubs, were also recorded, with probability of selection decreasing sharply at short distances (Fig 3). Probability of selection by capybaras increased with increasing NDVI during day- and nighttime periods, although the relatively probability of selection plateaued at a NDVI value of approximately 0.5 during nighttime periods.

**Table 2. Model selection across natural landscapes for day- and night periods, based on Akaike Information Criterion (AIC).** Table is ranked by ΔAIC.

| Model | K | AIC | ΔAIC | ω | $\chi^2$ |
|---|---|---|---|---|---|
| | | | *Natural landscapes (day)* | | |
| **Full** | **11** | **25887.1** | | **1** | **5700.4*** |
| Water | 5 | 27147.5 | 1260.4 | 0 | 4428.1* |
| Forest | 5 | 30365.4 | 4478.3 | 0 | 1210.1* |
| Open Areas | 5 | 30982.6 | 5095.4 | 0 | 593.0* |
| Null | 3 | 31571.6 | 5684.4 | 0 | |
| | | | *Natural landscapes (night)* | | |
| **Full** | **11** | **23411.9** | | **1** | **7598.6*** |
| Water | 5 | 24089.6 | 677.7 | 0 | 6908.9* |
| Open Areas | 5 | 30061.4 | 6649.5 | 0 | 937.1* |
| Forest | 5 | 30073.3 | 6661.4 | 0 | 925.3* |
| Null | 3 | 30994.6 | 7582.6 | 0 | |

Models with smaller AIC values were taken as the best to predict capybara habitat selection. Top-ranked model is highlighted in bold. Likelihood ratio test ($\chi^2$) is also displayed in table.

*p<0.001.

**Table 3. Capybara resource selection function coefficients (β) for both day- and nighttime across natural and human-modified landscapes.**

|  | *Natural landscapes* | | *Human-modified landscapes* | |
|---|---|---|---|---|
|  | *Day* | *Night* | *Day* | *Night* |
| Study Area* | 0.00 (0.00) | 0.00 (0.00) | 0.00 (0.00) | 0.01 (0.12) |
| Individual/Study Area* | 0.25 (0.51) | 0.21 (0.46) | 0.25 (0.50) | 0.06 (0.25) |
| NDVI | **0.21 (0.02)** | 0 (0.02) | **0.32 (0.02)** | 0 (0.02) |
| $(NDVI)^2$ | -0.01 (0.01) | **-0.03 (0)** | -0.01 (0.01) | **-0.15 (0.01)** |
| Forest Interior | **-0.63 (0.04)** | **-0.32 (0.04)** | **-0.83 (0.04)** | **-0.08 (0.03)** |
| (Forest Interior)$^2$ | **-0.8 (0.04)** | **-0.72 (0.04)** | **0.21 (0.01)** | **-0.04 (0.01)** |
| Grasses/Shrubs | **0.21 (0.05)** | 0.02 (0.04) | **1.03 (0.03)** | **0.57 (0.03)** |
| (Grasses/Shrubs)$^2$ | **-0.11 (0.02)** | **-0.02 (0.01)** | **-0.36 (0.02)** | **-0.39 (0.02)** |
| Water | **-1.52 (0.03)** | **-1.91 (0.03)** | **-0.84 (0.02)** | **-0.46 (0.02)** |
| (Water)$^2$ | **0.32 (0.02)** | **0.66 (0.02)** | **0.16 (0.01)** | -0.01 (0.02) |

Standard errors are displayed within the parentheses; Regression coefficients (β) with confidence intervals that did not overlap zero are highlighted in boldface.

*Random effects.

## Human-modified landscapes' models

Across human-modified landscapes, the full model was also top-ranked for both day- and nighttime periods (Table 4). Models strongly fit the data in these landscapes (*day average $r_s$ = 0.89; night average $r_s$ = 0.72*), with weaker results found in São Paulo municipality during nighttime, where capybaras were tracked in a non-agricultural state park (Table A in S3 Appendix). The most important variable in predicting capybara habitat selection for day- and nighttime periods was distance to grasses/shrubs (day: β = 1.03±0.03; night: β = 0.57±0.03; Table 3). Distance to water (day: β = −0.84±0.02; night: β = −0.46±0.02; Table 3) and distance to forest interior (day: β = −0.83±0.04; night: β = −0.08±0.03; Table 3) were also significant in predicting capybara habitat selection, with stronger coefficients found for daytime periods. NDVI was a weaker variable in predicting capybara habitat selection during daytime periods, when compared to other habitat variables, and was not significant during nighttime periods (day: β = 0.32±0.02; night: β = 0±0.02; Table 3).

Contrasting to natural landscapes, capybaras across human-modified landscapes were observed with higher preferences for forest interior areas and areas close to forests, with probability of selection declining with increasing distance to forested habitats (Fig 2). Capybaras also showed preferences for areas near water sources, with higher selection during the day (Fig 2). Lower preferences for areas close to grasses/shrubs were found for human-modified landscapes when compared to natural landscapes, with selection increasing at mid distances (125-m) and declining at larger distances (250-m; Fig 3). Similar to natural landscapes, the relative probability of selection increased with increasing NDVI values during daytime periods (maximum coefficients at NDVI values close to 0.7). For nighttime periods, the relative probability of selection peaked at a NDVI value close to 0.5.

## Discussion

This is the first study using GPS tracking, high-resolution imagery and resource selection functions (RSF) to analyze and quantify capybara habitat selection strategies across natural and human-modified landscapes. Capybaras strongly selected forested habitats across human-modified landscapes during daytime periods, whereas selection for forests was weak across

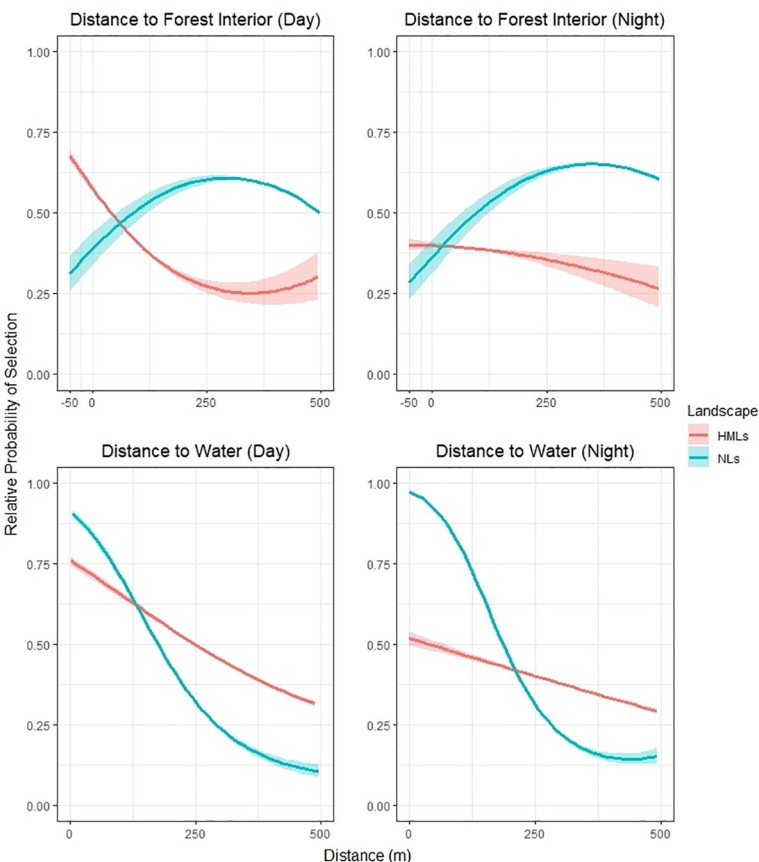

**Fig 2. Relative probability of selection of distance to forest interior and distance to water across natural and human-modified landscapes during day- and night periods.** The y axis represents the relative probability of selection, ranging from 0 to 1. The x axis represents distance to the habitat. Negative values of forest graphs are related to areas into the forest interior (-50m represents areas 50m inside forest patches).

both day- and nighttime in natural landscapes. This pattern of forest selection in human-modified landscapes may be a direct response to human activities (e.g. agricultural machinery, people and vehicle traffic), which are more intense in open areas of our study sites during daytime periods. As wildlife respond to human disturbance following the same principles used by prey encountering predators [60], capybaras may increase their selection for forests during daytime to avoid contact with humans. Indeed, other studies have suggested that forest cover may provide protection for capybaras from hunting [19], and capybara groups were observed seeking shelter in forests when humans approached (personal observation). Also, distance to the nearest riparian forest patch had a great influence in capybara habitat selection across human-modified landscapes of the Colombian Llanos [61].

The high selection for forests by capybaras across human-modified landscapes may put these amplifying hosts in closer contact with *A. sculptum* ticks, the main vector for the BSF agent, *R. rickettsii* [10], since degraded forests are the preferred habitats of *A. sculptum* ticks [24, 25]. A parallel study that evaluated same capybara groups of the present study reported an overall mean abundance of *A. sculptum* ticks on capybaras significantly higher across human-modified landscapes than in natural areas [37]. In addition, the environmental density of all host-questing stages of *A. sculptum* (larvae, nymphs and adults) was also significantly higher across human-modified landscapes than in natural landscapes [37]. Therefore, capybaras may

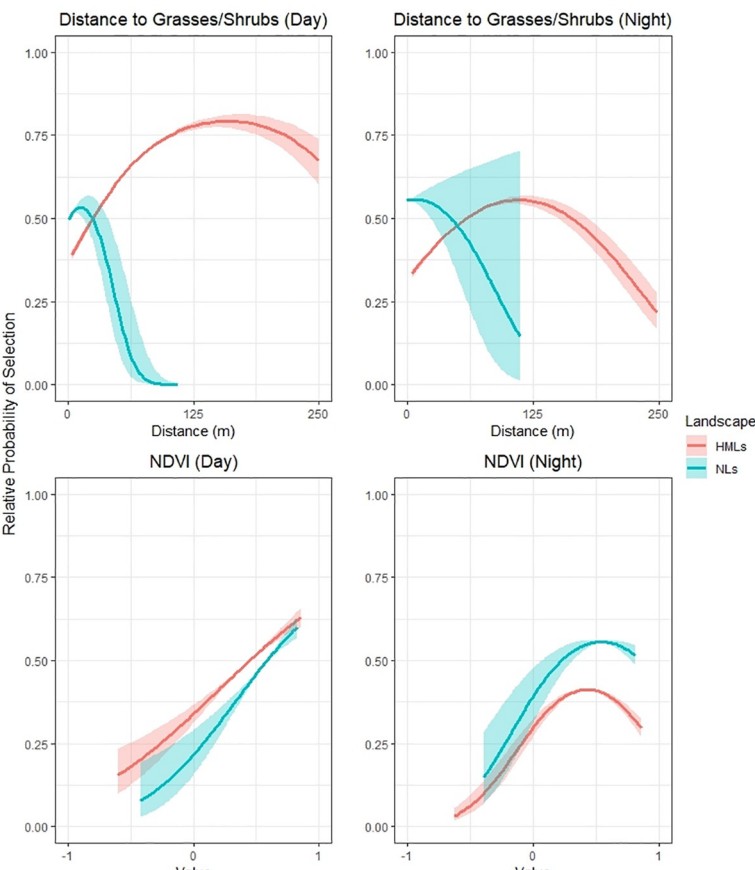

**Fig 3. Relative probability of selection for distance to grasses/shrubs and NDVI across natural landscapes and human-modified landscapes during day- and night periods.** The y axis represents the relative probability of selection, ranging from 0 to 1. The x axis represents the distance to grasses/shrubs or NDVI values.

be highly efficient hosts across human-modified landscapes, increasing their already described capacity in maintaining and carrying large numbers of *A. sculptum* [10], due to shared preferences for forested habitats with this tick [24, 25].

The ecological relationships between capybaras and *A. sculptum* are a key point in BSF epidemiology, since *A. sculptum* populations are not able to sustain *R. rickettsii* for successive generations without vertebrate-amplifying hosts [62, 63], among which capybaras stands out [10]. Capybaras are linked to the amplification of rickettsial infection among *A. sculptum* populations, creating new cohorts of infected ticks during bacteremia periods (days or weeks), when they maintain *R. rickettsii* in their bloodstream [10]. Consequently, minimizing the exposure of capybaras to *A. sculptum* reduce the populations of this tick, since capybaras are major hosts for *A. sculptum* [10]. Actions resulting in a drastic reduction of *A. sculptum* populations across our study areas are likely to limit *R. rickettsii* infection from tick populations, preventing new BSF cases [37].

Preferences for areas nearby water sources across natural and human-modified landscapes were not surprising. Capybaras are semi-aquatic mammals and their dependence on water sources has already been well-documented, with some authors reporting these rodents hardly moving more than 500-m from water [61, 64, 65]. However, our models highlighted that capybaras were less dependent on water sources in human-modified landscapes, which may be

**Table 4. Model selection across human-modified landscapes for day- and night periods, based on Akaike Information Criterion (AIC).** Table is ranked by ΔAIC.

| Model | K | AIC | ΔAIC | ω | $\chi^2$ |
|---|---|---|---|---|---|
| | | | *Human-modified landscapes (day)* | | |
| **Full** | **11** | **40628.2** | | **1** | **6678.9**[*] |
| Open Areas | 5 | 43203.9 | 2575.7 | 0 | 4091.2[*] |
| Forest | 5 | 43675.1 | 3046.9 | 0 | 3620.0[*] |
| Water | 5 | 45905.3 | 5277.1 | 0 | 1389.8[*] |
| Null | 3 | 47291.1 | 6662.9 | 0 | |
| | | | *Human-modified landscapes (night)* | | |
| **Full** | **11** | **44548.5** | | **1** | **259.5**[*] |
| Open Areas | 5 | 45396.3 | 847.8 | 0 | 847.5[*] |
| Water | 5 | 45571.3 | 1022.8 | 0 | 672.5[*] |
| Forest | 5 | 45984.3 | 1435.8 | 0 | 259.5[*] |
| Null | 3 | 46239.8 | 1691.3 | 0 | |

Models with smaller AIC values were taken as the best to predict capybara habitat selection. Top-ranked model is highlighted in bold. Likelihood ratio test ($\chi^2$) is also displayed in table.

[*] p<0.001.

related to human-driven variation in one or more behaviors linked to water use: reproduction, thermoregulation, or predator avoidance [12].

Quality and quantity of food resources from highly nutritious agricultural and pasture fields seems to have a strong influence on habitat selection by capybaras, since grasses/shrubs was the strongest variable in our human-modified landscapes' models. Because we wanted to compare selection for similar habitats across natural and human-modified landscapes, we did not separate crops and pastures into individual habitat classes. However, in the future, more detailed habitat selection studies for capybaras might consider fine-scale spatiotemporal dynamics of agriculture and pasture fields in human-modified landscapes. Understanding selection for these resources, mainly sugar cane, which is linked to BSF spread [66], may be essential to develop conflict mitigation strategies for the species.

Lastly, improving NDVI temporal resolution could potentially increase the link between this vegetation index and capybaras, since this variable was weak in predicting capybara habitat selection. Higher temporal resolution of NDVI may allow for further investigations on the interaction between vegetation quality and capybara habitat use.

Despite the small number of studied animals in the Brazilian Pantanal, capybaras in this area were tracked for relatively long periods with high numbers of GPS-locations, which increases data reliability. Our results showed clear distinctions between habitat selection of capybaras in natural and human-modified landscapes, providing a background for further investigation into the potential indirect effects of human disturbance in capybara space use. The development of knowledge regarding these effects may assist future management actions aimed at reducing conflicts linked to the species, and the exposure of capybaras to *A. sculptum* ticks.

## Conclusions

Through the use of GPS tracking and resource selection functions it was possible to demonstrate variation in habitat selection strategies of capybaras across natural and human-modified landscapes. Areas close to forested habitats were more selected with higher levels of probability across human-modified landscapes than across natural landscapes. In addition, capybaras

consistently selected areas near water in both landscapes, but this resource was more important in predicting capybara habitat selection in natural landscapes. In contrast, grasses/shrubs (which includes crops and pasture fields) was a stronger predictor of capybara habitat selection across human-modified landscapes. Our results show the influence of anthropic disturbance in capybara space use patterns and indicate that an increased understanding of capybara habitat use in natural and human-modified landscapes may support improved human-wildlife conflict management.

## Supporting information

**S1 Appendix. Land cover classification of capybara habitats.** Methods on how habitats of studied capybaras were classified using high-resolution satellite imagery and random forest algorithm.
(DOCX)

**S2 Appendix. Sensitivity analysis performed for study areas across natural and human-modified landscapes.** We performed sensitivity analysis to set the number of random points per 'use' point to our habitat selection models.
(DOCX)

**S3 Appendix. Top-ranked models' evaluation.** We used presence-only data to evaluate our top-ranked models' performance through Spearman rank correlations between area-adjusted frequencies and resource selection functions spatial bins.
(DOCX)

**S4 Appendix. GPS-locations of tracked capybara groups in natural landscapes of the Brazilian Pantanal and human-modified landscapes of São Paulo state, Brazil.** We plotted diurnal and nocturnal capybara locations at each study area over WorldView-2 satellite imagery.
(DOCX)

**S1 Table. Summary table for GPS-tracked capybaras across natural and human-modified landscapes.**
(DOCX)

**S1 Data.**
(CSV)

## Acknowledgments

We are grateful to SESC Pantanal, UFMT, Embrapa Pantanal and Alegria and São José ranches (Corumbá) for logistical support during field work in the Brazilian Pantanal and to the "Departamento de Água e Esgoto de Americana (DAE)" for allowing us to work at the "Estação de Tratamento de Esgoto (ETE) de Carioba" in Americana municipality. Forestry Institute and Coordination of Urban Parks of the São Paulo State Secretariat for the Environment offered logistical support in captures and monitoring (SMA Process 000.409/2015). We are also very grateful to GIS Lab at Smithsonian Institution, VA, USA, for the technical training in GIS and spatial analysis.

## Author Contributions

**Conceptualization:** Thiago C. Dias, Jared A. Stabach, Qiongyu Huang, Marcelo B. Labruna, Peter Leimgruber, Katia M. P. M. B. Ferraz, Beatriz Lopes, Ana M. Nievas, Patrícia F. Monticelli, Vlamir J. Rocha.

**Data curation:** Thiago C. Dias, Marcelo B. Labruna.

**Formal analysis:** Thiago C. Dias, Jared A. Stabach, Qiongyu Huang, Peter Leimgruber.

**Funding acquisition:** Marcelo B. Labruna.

**Investigation:** Thiago C. Dias, Marcelo B. Labruna, Katia M. P. M. B. Ferraz, Beatriz Lopes, Hermes R. Luz, Francisco B. Costa, Hector R. Benatti, Lucas R. Correa, Ana M. Nievas, Patrícia F. Monticelli, Ubiratan Piovezan, Matias P. J. Szabó, Daniel M. Aguiar, José Brites-Neto, Marcio Port-Carvalho, Vlamir J. Rocha.

**Methodology:** Thiago C. Dias, Jared A. Stabach, Qiongyu Huang, Marcelo B. Labruna, Peter Leimgruber, Katia M. P. M. B. Ferraz, Vlamir J. Rocha.

**Project administration:** Thiago C. Dias, Marcelo B. Labruna, Vlamir J. Rocha.

**Resources:** Marcelo B. Labruna, Peter Leimgruber, Vlamir J. Rocha.

**Supervision:** Jared A. Stabach, Qiongyu Huang, Marcelo B. Labruna, Peter Leimgruber, Vlamir J. Rocha.

**Validation:** Thiago C. Dias, Jared A. Stabach, Qiongyu Huang, Vlamir J. Rocha.

**Visualization:** Thiago C. Dias, Jared A. Stabach, Qiongyu Huang.

**Writing – original draft:** Thiago C. Dias, Vlamir J. Rocha.

**Writing – review & editing:** Jared A. Stabach, Qiongyu Huang, Marcelo B. Labruna, Peter Leimgruber, Katia M. P. M. B. Ferraz, Beatriz Lopes, Hermes R. Luz, Francisco B. Costa, Hector R. Benatti, Lucas R. Correa, Ana M. Nievas, Patrícia F. Monticelli, Ubiratan Piovezan, Matias P. J. Szabó, Daniel M. Aguiar, José Brites-Neto, Marcio Port-Carvalho, Vlamir J. Rocha.

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
