## [Decision Letter · Decision Letter 0]

20 Mar 2020

PONE-D-20-02991

Human-induced changes in habitat preference by capybaras (*Hydrochoerus hydrochaeris*) and their potential effect on zoonotic disease transmission

PLOS ONE

Dear Mr. Dias,

Thank you for submitting your manuscript to PLOS ONE. After careful consideration, we feel that it has merit but does not fully meet PLOS ONE’s publication criteria as it currently stands. Therefore, we invite you to submit a revised version of the manuscript that addresses the points raised during the review process.

This manuscript has now been reviewed by three experts in mammal responses to human activity, movement ecology, and wildlife disease. All reviewers agree that the manuscript addresses an interesting and important topic and that the methods are generally robust. A consistent comment across all three reviews is that the authors have not explicitly linked changes in habitat preference to the spread of BSF (nor measured infection) and that the title and some of the conclusions should be altered. The reviewers otherwise provide various suggestions to improve the clarity and structure of the manuscript, including conceptualization in the Intro and the presentation of results. 

We would appreciate receiving your revised manuscript by May 04 2020 11:59PM. To enhance the reproducibility of your results, we recommend that if applicable you deposit your laboratory protocols in protocols.io, where a protocol can be assigned its own identifier (DOI) such that it can be cited independently in the future. For instructions see: http://journals.plos.org/plosone/s/submission-guidelines#loc-laboratory-protocols

We look forward to receiving your revised manuscript.

Kind regards,

Daniel Becker

Academic Editor

PLOS ONE

Journal Requirements:

2. We note that Figure 1 in your submission contain map images which may be copyrighted. All PLOS content is published under the Creative Commons Attribution License (CC BY 4.0), which means that the manuscript, images, and Supporting Information files will be freely available online, and any third party is permitted to access, download, copy, distribute, and use these materials in any way, even commercially, with proper attribution. For these reasons, we cannot publish previously copyrighted maps or satellite images created using proprietary data, such as Google software (Google Maps, Street View, and Earth). For more information, see our copyright guidelines: http://journals.plos.org/plosone/s/licenses-and-copyright.

Reviewers' comments:

Reviewer's Responses to Questions

**Comments to the Author**

1. Is the manuscript technically sound, and do the data support the conclusions?

Reviewer #1: Partly

Reviewer #2: Yes

Reviewer #3: Yes

2. Has the statistical analysis been performed appropriately and rigorously? 

Reviewer #1: Yes

Reviewer #2: Yes

Reviewer #3: Yes

3. Have the authors made all data underlying the findings in their manuscript fully available?

Reviewer #1: Yes

Reviewer #2: Yes

Reviewer #3: Yes

4. Is the manuscript presented in an intelligible fashion and written in standard English?

Reviewer #1: Yes

Reviewer #2: Yes

Reviewer #3: Yes

5. Review Comments to the Author

Reviewer #1: Human-induced changes in habitat preference by capybaras (Hydrochoerus hydrochaeris) and their potential effect on zoonotic disease transmission

Thank you for the opportunity to review this paper on capybaras and their habitat use in natural and human-modified settings. I found the paper overall interesting to read, but believe it would benefit from further detail in the introduction and some restructuring of results. It also has a number of small grammatical errors that should be fixed before resubmission.

Major comments:

1. Lines 66-68, sentence beginning “Across these landscapes”: This sentence could use some elaboration as it touches on several large concepts (human disturbance, wildlife responses, zoonotic disease transmission). Putting in some concrete examples may help here.

2. Please provide more detail on the second paragraph of the introduction. For example, it isn’t mentioned that capybaras are found in South America, and it isn’t clear if their populations are increasing across all of their distribution or just certain places.

3. Lines 85-87: Please expand on this sentence—how might human-driven variation in capybara habitat selection affect BSF dynamics? Could you provide more details from References 16-19 for readers who aren’t as familiar with the previous work that has been done?

4. Table 1: Was there a reason that there wasn’t a fifth model with just NDVI and NDVI^2?

5. Tables 2 and 4: Given that the full model was by far the best for natural and human-modified habitats, in both day and night, I think these tables could be moved to the supplementary material. It would also be preferable to rank the models by delta AIC.

6. Table 3: This might be easier to interpret as a figure (e.g. forest plot) rather than a table.

7. Lines 353-354: Are there direct observations of capybaras moving to avoid people, traffic, etc?

8. Lines 372-374: I found that this conclusion was not supported by the data. Given that there was no sampling of ticks and R. rickettsia from the capybaras, I don’t think the authors can conclude definitively that human-driven changes in capybara habitat selection have influenced BSF spread. Providing more details from previous work such as Reference 19 could be helpful here. The authors could also qualify their language to clarify that they did not directly study the role of capybaras in BSF spread.

9. The second to last paragraph of the Discussion felt like it did not add new information, and might belong more in the Introduction.

Minor comments: There were a number of small grammatical errors. It may help the authors to have another native English speaker read over the paper.

1. Can refer to just BSF rather than “the” BSF. e.g. lines 49, 50, 391

2. Line 49: insert “it is” before essential

3. Line 65: change “is” to “are”

4. Line 69: change “obtain” to “obtaining”

5. Line 77: remove “a” before rapid and change “grow” to “growth”

6. Line 81: change “rickettsioses” to “rickettsiosis”

7. Line 86: change “have” to “has”

8. Line 102: remove unnecessary space in “Faculty”

9. Line 110: please provide an overview sentence describing what HFI is

10. Lines 122-124: Since HFI incorporates human population density, this seemed a little repetitive.

11. Line 136: change “support” to “supports”

12. Line 166: change “trap” to “traps”

13. Line 179-180: What was the original time interval, if data were rarified to a 4 hour interval?

14. Line 221-222: This sentence should have come earlier in the introduction, as it was unclear why day and nighttime periods would be different

15. Line 224: change “locations” to “location”

16. Line 296: change “overlapped” to “overlap”

17. Line 303: remove “has”

18. Line 359: add “the” before “preferred”

19. Line 384: add “capybaras” after “by”

20. Line 402: change “increase” to “increased”

Reviewer #2: Review for “Human-induced changes in habitat preference by capybaras (Hydrochoerus

hydrochaeris) and their potential effect on zoonotic disease transmission”

This study compares the habitat selection preferences of capybaras in Brazil between human-modified and natural landscapes and across day vs. night periods. They contextualize their results with respect to risk of capybaras encountering Amblyomma sculptum in forested habitat patches and translocating them, increasing human exposure risk for Brazilian Spotted Fever. The authors use sophisticated and generally well-justified methods to compare candidate models representing different habitat types. I have a few comments to strengthen the rationale and improve the interpretation of results.

Major comments

Title: I think that the wording “and their potential effect on zoonotic disease transmission” implies that you measured something pertaining to pathogen transmission. Something like “Habitat selection in human-modified landscapes by capybaras, an important host for Amblyomma sculptum ticks” would be more accurate to the contents of the analysis

Introduction

L85-87: I would suggest adding more support for how capybara habitat selection in human modified areas would impact BSF epidemiology. What shifts in habitat selection would increase human risks?

Methods

Figure 1: Does the gray shading in rightmost panels indicate topography? I think it would be more useful to show the human footprint layer or land cover data instead.

L180: what was your definition of diurnal and nocturnal locations based on the time on sunrise and sunset? Did a location have to be within a certain number of hours after sunrise and before sunset to be considered diurnal?

Results

It would be helpful to see a map figure with capybara locations on it.

Table 3: please report the random effect variance

Discussion

L353: could the increased selection for forested areas be also simply a function of lower forest availability in human modified landscapes? Highlighting that capybaras selected more strongly for forested areas during the day vs the night when capybaras in natural landscapes did not show that difference with time period would strengthen your argument.

L355: this is not clear. What do you mean by the relationship being more pronounced in open areas? Was the location of the animal within the study area taken into account?

L359 – 364: this information about A. sculptum habitat preferences is important for your rationale and should be in the introduction

I think there should be some caveats in the discussion section acknowledging the small sample size for capybaras in the natural landscapes. It is difficult to generalize based on 4 individuals.

L416: Did you compare the use of forested areas between these landscapes? I thought the interpretation would be that capybaras selected for areas closer to forests in human modified landscapes than natural landscapes.

Minor comments

L68: should be “obtaining”

L77: “recently experiencing a rapid grow” should be rephrased to something like “recently experiencing rapid population growth in modified habitats”

L91: remove “must”

L102: remove space in “Faculty”

L160: were these cattle ranches?

L223-225: it could be clearer that the random points were generated within the buffers. So did each capybara have a different buffer size?

L236: change “cross-time” to “across time”

L367: citation needed

L384: should be “…have a strong influence on habitat selection by capybaras, since…”

Reviewer #3: Minor revisions

L47: I would be more careful when making the statement that capybara is a “conflict species”, please reword. You can say of course that the species can cause problems in some areas (which ones), but such an overarching adjective is not appropriate for any species. Besides, the argument after the sentence makes me think that you are talking about a different kind of issue, the Human Wildlife Conflict (HWC) is usually well define, please revise this and reword accordingly.

L49: all cases around the world? The capybara is only Neotropical, even other regions issues are caused by capybara? What is “has been implicated”, can you be more specific?

L54: higher?

L57: can you provide the direction of the effect of the most important variables? (is water = distance to water? )

*the abstract is focus on the resource selection and we don´t really understand the link with the disease transmission which is included as a main topic in the title, can you comment on that, would be great.

L65: just in case, not all species are forced to adapt to HDL, actually many others also show preferences and look for human infrastructures, but not the topic of the paper…

L67: spatial ecology is too broad, and actually makes us think about the discipline, can you be more specific, e.g. distribution, occupancy, habitat use?

L81. Lethal to who? Cattle, sheep, horses? All domestics?

L90. It just sounds weird to put the wildebeest to compare capybaras, they are completely different species/groups, I would change the ref

L91. The first prediction does not really sound like a prediction, please be more specific of what you expect (e.g. how variation?), it seems that it is contained in b so maybe there is no A? (please reword “must show” for something more formal)

L105. 11 groups mean you tracked all members of each group?, please rephrase accordingly, would be good to have an indication of the number of individual per group, that also tends to affect animal movements. I don´t think this part fits well into “study area” and rather confuses the reader. It sounds more like an analysis or any other section. Please move accordingly and focus just on the area itself, the variables can go in other section.

L110: please clarify (in general) how this index works

L118: so you did tag several individual within each group? Then you get a mean dispersal distance for the group, and you use that area to calculate the HFI? When do you sum up the values. This part is a bit confusing, please reword. Why summing is better than averaging? Please justify so the reader understands better (for what is stated in L20 it seems to be a mean actually)

Please double check the English and grammar in a few places.

Lastly, I know the main purpose of the paper was not to study the transmission of the disease itself, but I found it difficult to link the actual study (RSF with capybaras) to the spread of the BSF. Although the arguments are strong and well supported by references, it would have been great if you had quantified something from the animals that were captured. If you have more information, that would make the suggestions stronger (for example, an estimation of the actual number of infested animals in your groups), it would contribute to make the hypothesis more attractive.

Best wishes, I look forward to seeing this paper published.

6. PLOS authors have the option to publish the peer review history of their article (what does this mean?). If published, this will include your full peer review and any attached files.

Reviewer #1: No

Reviewer #2: No

Reviewer #3: Yes: Lain E. Pardo

---

## [Author Response · Author response to Decision Letter 0]

17 Jun 2020

Reviewer #1: Excluding that we decided to keep Tables 2 and 4 in the main text, we have incorporated all of your suggestions in our manuscript. Thank you very much, they were very important to improve the quality of our manuscript.

Reviewer #2: Your suggestions were very helpful to us, we have incorporated all them in our revised manuscript. Thank you for suggesting a new title.

Reviewer #3: Thank you very much for all of your comments and suggestions, they were all incorporated in the revised manuscript.

---

## [Decision Letter · Decision Letter 1]

21 Jul 2020

PONE-D-20-02991R1

Habitat selection in natural and human-modified landscapes by capybaras (*Hydrochoerus hydrochaeris*), an important host for *Amblyomma sculptum* ticks.

PLOS ONE

Dear Dr. Dias,

Thank you for submitting your manuscript to PLOS ONE. After careful consideration, we feel that it has merit but does not fully meet PLOS ONE’s publication criteria as it currently stands. Therefore, we invite you to submit a revised version of the manuscript that addresses the points raised during the review process.

Thank you for the careful revision. Please see some final minor comments from one of the reviewers, primarily about Figure 1. However, I am confident the authors can address these. 

We look forward to receiving your revised manuscript.

Kind regards,

Daniel Becker

Academic Editor

PLOS ONE

Reviewers' comments:

Reviewer's Responses to Questions

**Comments to the Author**

1. If the authors have adequately addressed your comments raised in a previous round of review and you feel that this manuscript is now acceptable for publication, you may indicate that here to bypass the “Comments to the Author” section, enter your conflict of interest statement in the “Confidential to Editor” section, and submit your "Accept" recommendation.

Reviewer #1: All comments have been addressed

Reviewer #3: All comments have been addressed

2. Is the manuscript technically sound, and do the data support the conclusions?

Reviewer #1: Yes

Reviewer #3: Yes

3. Has the statistical analysis been performed appropriately and rigorously? 

Reviewer #1: Yes

Reviewer #3: Yes

4. Have the authors made all data underlying the findings in their manuscript fully available?

Reviewer #1: Yes

Reviewer #3: Yes

5. Is the manuscript presented in an intelligible fashion and written in standard English?

Reviewer #1: Yes

Reviewer #3: (No Response)

6. Review Comments to the Author

Reviewer #1: (No Response)

Reviewer #3: Dear authors,

I think the paper is clearer now.

The only major comment is related to the map and the structure of the study area.

I am confused about figure 1. In the first manuscript you provided a much better figure. The current is not in a good resolution, but most importantly it is not informative. Please use the first figure for publication. Over this map, I would highly recommend putting a landuse/cover layer on the orange area (HDL), as it would give us a clear picture of the patchiness and landuses across that area. The green section must be just pantanal, but would be good to see the other elements included under this category (e.g. riparian forest). Further you are providing too much details about how you constructed the map (e.g. using R etc), that should stay in the methods section. Not needed for the caption. Would it be possible to include the locations of the animals at each location? (would be a very nice supplementary data).

In the study area section there is still a lot of information that is more concerned to the analysis itself. For example, the paragraph describing how they assess the level of human disturbance must be moved to the adequate position in analysis. In study area just provide the geographical context and landscape characteristics so we understand what is human dominated vs natural etc. (paragraph of L138 is also data anaysis, when you move it, please state at the beginning why you did what you are describing e.g. “to xxx we calculated the mean….as currently stants it is not very clear, e.g. so the index also need the GPS coordinates then? Seems like but if the reader is told at the beginning of the paragraph then it would make more sense. Move parag L145 to the end.

Minor comments,

In L140. Don´t say “found by colleagues” just cite the reference.

L157 “what is impressive” I would avoid this adjective, sounds rather subjective

For the discussion it might be worth mentioning the results of Pardo et al. 2020 (Land Management strategies can increase oil palm use by mammals in Colombia; https://www.nature.com/articles/s41598-019-44288-y). They found that “habitat use was 3.6 times greater in riparian forest than in oil palm, and decreased with increasing distance to nearest forest patch, with detections never occurring further than 357 m from remnant forest”. This supports your findings of higher preferences for forest interior in HDL and areas close to forests. This could help to support your personal observation in L399 or L425 or elsewhere.

I look forward to seeing this published. Congratulations.

Best

• I can´t confirm authors made all data underlying the findings in their manuscript fully available. This is responsibility of the authors.

7. PLOS authors have the option to publish the peer review history of their article (what does this mean?). If published, this will include your full peer review and any attached files.

Reviewer #1: No

Reviewer #3: No

---

## [Author Response · Author response to Decision Letter 1]

31 Jul 2020

Reviewer #3

Thank you for your new suggestions, including adding the results of Pardo et al. 2020 in our discussion. We made all changes you requested and are submitting our paper with a new Figure 1, using a land cover basemap.

---

## [Editor Report · Decision Letter 2]

7 Aug 2020

Habitat selection in natural and human-modified landscapes by capybaras (*Hydrochoerus hydrochaeris*), an important host for *Amblyomma sculptum* ticks.

PONE-D-20-02991R2

Dear Dr. Dias,

We’re pleased to inform you that your manuscript has been judged scientifically suitable for publication and will be formally accepted for publication once it meets all outstanding technical requirements.

Kind regards,

Daniel Becker

Academic Editor

PLOS ONE
---

## [Editor Report · Acceptance letter]

11 Aug 2020

PONE-D-20-02991R2 

Habitat selection in natural and human-modified landscapes by capybaras *(Hydrochoerus hydrochaeris)*, an important host for *Amblyomma sculptum* ticks. 

Dear Dr. Dias:

I'm pleased to inform you that your manuscript has been deemed suitable for publication in PLOS ONE. Congratulations! Your manuscript is now with our production department. 

Kind regards, 

on behalf of

Dr. Daniel Becker 

Academic Editor

PLOS ONE